# The Experiences of People Living with Peripheral Neuropathy in Kuwait—A Process Map of the Patient Journey

**DOI:** 10.3390/pharmacy7030127

**Published:** 2019-08-30

**Authors:** Maryam Alkandari, Kath Ryan, Amelia Hollywood

**Affiliations:** School of Pharmacy, University of Reading, Reading, RG6 6AP, UK

**Keywords:** peripheral neuropathic pain, diabetes, patient experiences, process map, Kuwait healthcare

## Abstract

Peripheral neuropathy is a neurological disease characterised by pain, numbness, tingling, swelling or muscle weakness due to nerve damage, caused by multiple factors such as trauma, infections and metabolic diseases such as diabetes. In Kuwait 54% of the diabetic population, has peripheral neuropathy. In this exploratory, qualitative study conducted in Kuwait, 25 subjects with peripheral neuropathy took part in one-on-one, semi-structured interviews lasting 45–60 min. Interviews were transcribed, translated into English and coded using NVivo 12. Four individual patient journeys were mapped out in detail, then compared and condensed into a single process map. The remaining 21 interviews were then reviewed to ensure the final map represented all patient journeys. Participants reported similar healthcare pathways for their peripheral neuropathy and faced various difficulties including lack of psychological support, administrative issues (long waiting referral periods, loss of medical documents, shortage of specialists and lack of centralized electronic medical records) and inadequate medical care (shortage of new treatments and deficient follow-ups). Mapping the patient journey in Kuwait showed similar pharmacological treatment to UK guidelines, except that some medicines were unavailable. The map also indicated the need for an integrated referral approach, the use of technology for electronic medical recording and report transmission, alongside education on self-management, coping mechanisms and treatment options for people living with peripheral neuropathy.

## 1. Introduction

Peripheral neuropathy is a common neurological disorder affecting people from both developed and developing countries. It can be caused by various conditions such as vitamin deficiency, traumatic injury, alcoholism, immune system diseases, viral infections or most commonly diabetes [1,2,3]. In 74–82% of people, the cause can be determined by a combined approach of patient medication history, examinations and ancillary testing [4]. Symptoms of peripheral neuropathy include sensory symptoms (e.g., numbness and tingling), weakness, autonomic symptoms (e.g., impotence, orthostatic hypotension and sweating abnormalities) or neuropathic pain (burning, stabbing, electrical) [5]. Kuwait is a high-income country providing a high standard of health and social services to its citizens with health indicators similar to those of highly developed countries [6]. The most common causes of peripheral neuropathy in Kuwait are diabetes and lower back pain as discussed below [7]. Furthermore, the incidence of diabetes in the adult population in Kuwait is one of the top five in the world, with 18% of the population (424,000 in 2019) diagnosed as diabetic. This number has increased dramatically in the last decade [8].

The prevalence of peripheral neuropathy in the general population (worldwide) is 2.4% and increases to an estimated 8% in those older than 55 years [9,10]. In Western societies, the most common cause is diabetes mellitus, with a prevalence of peripheral pain in diabetics ranging from 30–66% [11,12,13,14]. In a population-based study in the Netherlands diabetes was found to be the third main cause of peripheral neuropathy [15]. Hall et al. [16] found that there were approximately 15.3 cases of neuropathic pain for every 100,000 individuals in the United Kingdom between 1992 and 2002 and since then this ratio has been increasing every year. In Kuwait, in 2010, peripheral neuropathic pain was estimated to affect around 39% of the diabetic population [17]. Like elsewhere in the world, the incidence and prevalence of diabetes also increases dramatically with age in Kuwait, which indicates a rise in the number of people who will have to live with peripheral neuropathy [18]. In addition, lower back pain is prevalent among Kuwaitis, which leads to peripheral neuropathy. In Kuwait, the prevalence of lower back pain among 10-18 year-olds was found to be 58% (51% in males and 65% in females), increasing with age in both males and females, thus increasing the risk for developing peripheral neuropathy [19].

In western countries, multidisciplinary care for people living with peripheral neuropathy is encouraged and receiving a wide range of additional support allows people to manage their own condition [20]. Most research conducted in the Arab region has only reported the prevalence of peripheral neuropathy [21,22] and research focusing on patients’ experiences in the Middle East is lacking. The main purpose of this research was to explore the experiences of people living with peripheral neuropathy by examining the healthcare pathway in Kuwait from the patient perspective. The healthcare pathway will be examined in terms of referrals, investigations and general management. This paper uses the technique of process mapping to chart the journey experienced by people living with peripheral neuropathy in Kuwait and to compare that journey with standards existing in peripheral neuropathy care in Western countries. Process mapping allowed us to see and understand the patient’s experience [23] by separating the management of peripheral neuropathy and its treatment into a series of consecutive events or steps (for example, activities, interventions or interactions with healthcare professionals).

## 2. Materials and Methods

### 2.1. Study Design

This qualitative study aimed to explore the pathway of standard care for the management of people living with peripheral neuropathy in Kuwait and to develop a schematic process map of the patient journey based on their experiences. The process map was then compared with existing guidelines in the UK, where the National Institute for Health and Care Excellence (NICE) and the International Association for the Study of Pain (IASP) guidelines are used.

### 2.2. Study Setting

The study was conducted at Ibn Sina Neurology and Neurosurgery Hospital, the tertiary centre to which neurology patients are referred by neurology specialists in general hospitals. The principal investigator informally observed people living with peripheral neuropathy that attended the outpatient clinic of Ibn Sina Hospital, to familiarise themselves with the healthcare setting.

### 2.3. Recruitment Strategy

A study summary was provided to the neurologists attending the outpatient clinics to identify potential participants meeting the eligibility criteria (aged over 18 years, diagnosed with peripheral neuropathy, resident in the State of Kuwait, speaking Arabic or English). The nursing staff distributed an information pack (consisting of an invitation letter, patient information sheet and pamphlet with further contact details) to potential participants. Interested people contacted the principal investigator via phone or email.

Recruitment started in February 2017 and was completed within two weeks. Of ninety-five potential participants, twenty-seven contacted the principal investigator (28% response rate). After obtaining consent, the principal investigator undertook an in-depth review of the medical records of these outpatients to confirm that they met all the inclusion criteria. Two people were excluded because one did not meet the age criteria (was under 18 years old) and the other was excluded on the grounds of having neurological problems other than peripheral neuropathic pain. Therefore, twenty-five people (20 Kuwaiti and 5 Non-Kuwaiti) were deemed eligible for inclusion in the study.

### 2.4. Data Collection

A semi-structured interview guide was used to explore people’s experiences. Participants were asked initial demographic questions, including age, sex and nationality, along with questions regarding comorbidities and duration of peripheral neuropathy (see Table 1). The interview proceeded with open-ended questions that began by broadly asking about their pain and then moved on to more specific questions about the healthcare they received in terms of medical treatment, medication and whether they felt anything was missing from their care. The interview guide was developed from the literature, taking into consideration the culture and healthcare system in Kuwait. The healthcare system in Kuwait is comparable with Western processes in terms of referrals from primary care to hospital settings, however long referral times and a lack of specialist care, such as psychological services, are key differences.

Interviews began in March 2017 and were conducted over a period of a year. Participants were assigned study specific numbers and initials to anonymize their identity. Each participant was interviewed individually in a private room in the hospital, in the language of their choice (English or Arabic). Out of the five non-Kuwaiti participants, three preferred English. All Kuwaiti participants preferred Arabic. Interviews lasted 45–60 min and were audio-recorded. The English interviews were fully transcribed in English; the Arabic interviews were transcribed in Arabic and later translated into English. From the twenty-five transcripts, six were selected for review to ensure that the transcriptions and translations were accurate. Four transcripts were reviewed by an academic lecturer at the Languages Centre at the University of Jordan, who was proficient in both Arabic and English; and two by a bilingual physician at the Department of Community Medicine and Behavioural Sciences, Kuwait University. All reviews confirmed that the transcriptions and translations were accurate and consistent.

### 2.5. Data Analysis

The transcripts were transferred to NVivo12 software [24] for data management and analysis. The data was initially manually coded inductively after familiarization and then by a combination of text search queries and coding queries and then extracted to generate a preliminary report which helped to plot the patient journey process map, which in turn aided in identifying the weaknesses in the existing patient care pathway. It also helped to formulate suggestions for improvement, propelled by the implicit and explicit deductions made by the researchers. Validation was ensured not only by prolonged engagement but also by peer debriefing.

A planned approach for producing a process map was applied to the data to ensure a comprehensive representation of the patient journey [25]. Four transcripts were chosen based on their rich data (eloquence of interviewee, length of interview, severity of pain) to plot four individual pathways. These pathways were compared to identify similarities (such as referrals and pharmacological treatments) and differences (such as disparities in referral time and waiting periods). These were combined to make a preliminary process map which reflected even the minutest details for example, signs, symptoms, investigation results, physical examinations, referral systems, follow-ups and their current situation. This was expanded into a second version by layering on details from the twenty-one remaining transcripts, which involved adding further detail and clarification to the preliminary process map drafted from the initial four participants. After multiple iterations, a simplified and condensed process map (see Figure 1) was produced that captured the journey for all people living with peripheral neuropathy in Kuwait.

### 2.6. Ethical Consideration

Ethical approval was obtained from the University of Reading Ethics Committee (UREC–16/46) as well as the Standing Committee for Health and Medical Research Coordination (SCHMRC) in the Ministry of Health (MoH) in the State of Kuwait (Ref no: 194/2014).

## 3. Results

The primary focus of this paper is to explore people’s experiences with peripheral neuropathy and identify the care pathway in Kuwait. The results were derived from interviews with people attending the outpatient clinic of Ibn Sina Hospital. Most participants were Kuwaiti (n = 20), with an average age of 55 years (SD = 10) and had been living with peripheral neuropathy for an average of 14 years (SD = 7). Further details of the participant characteristics can be found in Table 1.

The main symptoms experienced by people living with peripheral neuropathy were neuromuscular; and their most frequent complaints were physical pain in the feet, hands, extremities, back, leg and knees, as well as numbness and tingling. Close analysis of the interviews showed that subjects experienced similar treatment and care pathways despite differences in the presentation of their complaints. The process map of the outpatient journey, as described by patients in Kuwait and as depicted in Figure 1, shows a referral from a local primary clinic to the general hospital, where people living with peripheral neuropathy would see a general doctor and a neurologist, then on to a specialist/consultant at the national neurology hospital.

### 3.1. Primary Care Clinic—the Journey Starts

Administrative issues at the primary clinic were strongly evident to most of the people living with peripheral neuropathy. There were usually only a few members of staff (doctors or nurses) on duty at the clinic and this led to a long queue of people and long waiting times.
Some people living with peripheral neuropathy feel more pain while sitting on cold seats, awaiting their turn to see the doctor—Female, 82 years old, living with peripheral neuropathy for 20 years.

Some people were able to bypass the long waiting time by using their social influence (connections) in healthcare. Kuwait also has a green card system that is intended to function as a waiver of such waiting periods, to ensure that old people, for example, have short or even no waiting times and can complete their hospital visits quickly. For some participants, however, this card did not seem to fulfil its function.
Once, I entered the staff room to tell them that they must let me enter and see the doctor fast, as I had a green card and the right to see the doctor fast…One of them replied that I should wait, as there is someone preceding me in the queue and there are others who suffer from rheumatism too—Female, 82 years old, living with peripheral neuropathy for 20 years.

To also cut the waiting time, some people relied on other bypass methods, such as giving bribes to the porters.
I had to wait a month and a half and I also needed Wasta (a gift or bribe) to get the magnetic resonance image—Male, 35 years old, living with peripheral neuropathy for 15 years.
The Bengali porter, who knows everyone inside the clinic, is given two or five Kuwaiti Dinars to help in getting a faster service, some others call their doctor friends and other acquaintances to help them avoid these queue systems—Male, 62 years old, living with peripheral neuropathy for 10 years.

Furthermore, participants thought that the care given by some doctors, inside the consultation rooms, did not meet their expectations.
There was a doctor who keeps speaking on the phone and does not care about my sick condition or doesn’t tell me to sit or not—Female, 82 years old, living with peripheral neuropathy for 20 years.

This participant commented that her Kuwaiti senior friends would rather use their health insurance to go to a private hospital. The Kuwaiti government levies around two percent of a public servant’s salary in taxes and retirees receive a health-card entitling them to free care in all private hospitals.

At the primary clinic, the doctor was engaged with the patient for less than five minutes, during which time their medical history was quickly assessed; a judgement was made about which laboratory tests to conduct, such as blood glucose (see Table 2) and the doctor reviewed the current treatment of symptoms. A variety of problems were reported from the patients’ perspective, including various areas of patient care and prescription difficulties.

Prescriptions usually included general pain management using basic analgesics (see Table 3).
Unfortunately, the doctors here are not very familiar with this matter but only give the painkillers, do not treat the disease and pain, do not tell me what this pain is and do not treat the inflammation—Female, 45 years, living with peripheral neuropathy for 9 years.

Depending on the duration of their symptoms, intensity of pain and response to the prescribed medication, the patient will obtain a standard or an emergency referral from their primary care general practitioner to the general hospital.
They give a medical appointment for three months or two or two and a half months and if you missed your medical appointment, it is a big problem—Female, 56 years old, living with peripheral neuropathy for 30 years.

Due to the long waiting periods, and sometimes feeling a lack of care from doctors, some patients exited mainstream care to try alternative therapies such as acupuncture, massage, sujok (a Korean method of reflexology) and herbal medicine.

### 3.2. General Hospital—Multiple Referrals

#### 3.2.1. Referral to Doctors in a General Hospital

Administrative problems during referral mean it can take anywhere between 2 and 8 weeks to get an appointment at the general hospital depending on the waiting list in the local area. Since Kuwait does not have a centralized electronic medical record system, the paper medical files or referral documents play an essential role in the care of patients. These documents might get lost or go missing from the clinic or the patient might lose their referral letter. In such cases, there are detrimental consequences. The patient may have no option but to repeat the whole process of opening a file and getting a referral letter, thus increasing the waiting period to as much as 12 weeks. This long waiting time usually tests the patience of the individual or causes them to seek alternative therapies.
My file was lost. When I went to the general hospital and showed them my referral paper, they said to go back to the clinic and reopen a new file and then come. It took me another three months to finally meet this doctor as I had to wait again—Male, 62 years old, living with peripheral neuropathy for 10 years.

Doctors in the general hospital have to repeat the entire process by conducting basic laboratory tests, such as HbA1c and other procedures, such as advanced neurological investigations, to ensure there is a full investigation. They then continue appropriate pharmacological treatment, often without adequate patient notes and lack of proper communication and cooperation with their peers. Furthermore, there are time pressures and a heavy patient load. Table 3 presents the medication prescribed for general pain management.
If you see a doctor, you find him either on leave or travelling and thus you have to go to another doctor who does not know anything about your condition. So, the two doctors are the same but the second one prescribes the medication and the diagnosis. Days are passing in my life and I still feel pain and loss at the same time—Female, 55 years old, living with peripheral neuropathy for 30 years.

Depending on factors such as their symptoms, response to medicine and social status people were prescribed medications and asked to return for follow-up after a few weeks or months. Some people did not visit the doctor for follow-up for various reasons, including the effort involved, enduring the waiting time and the possibility of administrative problems, for example, their record not being available. Other people preferred to send their relatives or a representative to the pharmacy to get a repeat prescription without even seeing the doctor. Prescriptions could be repeated numerous times, without a requirement to reassess the patient’s condition.
In Kuwait, doctors pay no attention to the nature of the pain. They only give the patient painkillers and never know the cause—Female, 45 years old, living with peripheral neuropathy for 9 years.

#### 3.2.2. Referral to Neurologist in General Hospital

If deemed necessary, people may also be referred by the doctor in the general hospital to a neurologist in the same hospital. There are only five general hospitals that provide healthcare to approximately 4 million people living in Kuwait. An appointment might take 3–6 months depending on the waiting list and may even extend for a further two months due to the loss of medical documents in between. Some patients felt that their doctor did not give them their full attention.
The doctor should know the medical record of the patient before the patient enters. Here, doctors turn to the computer or read the medical record during meeting the patient, rather than giving us their full attention—Male, 47 years old, living with peripheral neuropathy for 3 years.

The neurologist usually repeats the full investigations, as done by their predecessor but also adds advanced investigations along with conducting further neurology-specific tests, such as the assessment of degree of loss of protective senses and reflexes and continues appropriate pharmacological treatment as mentioned in Table 3. If the patient still shows no or minimal improvement, they will be referred to the national hospital for neurology treatment. People living with peripheral neuropathy also reported exiting at this point to try alternative medicine.

### 3.3. National Hospital for Neurology—Ultimate Destination for Peripheral Neuropathy Care

The delay in getting an appointment for referral to the national hospital for neurology, including the waiting period and possible extension time of 3–6 months, plus another 2 months in case of loss of medical documents, inevitably added to the difficulties caused by administrative problems. This led to weariness and frustration for most patients, although some received quicker referrals due to their social influence or high economic status.
Long referrals of up to a year! And pain still continues… I think if the whole subject was based on money, they would pay more attention—Female, 63 years old, living with peripheral neuropathy for 24 years.

The neurology specialist/consultant at the national hospital started from the beginning once again in terms of history taking, examinations and treatment. They conducted full laboratory investigations including HbA1c and a physical examination, placing specific emphasis on neurovascular examinations (see Table 2). Nerve conduction studies and advice on podiatry care for people with diabetic peripheral neuropathy were also provided here (16 of the 25 people interviewed, had type 2 diabetes mellitus). Again, depending on the level of their symptoms, as well as the progress of their disease or their response to medication, people could be further referred to another experienced neurology specialist/consultant in the same hospital.

At this stage, patients received a structured pharmacological regimen, similar to the UK guidelines for peripheral neuropathy pain care, consisting of anticonvulsants as the first line of treatment. If there was an inadequate response, the patient received the second line of treatment and if needed, there was an escalation to the third line of treatment, which included opioids. Table 3 reflects the pharmacological treatment options that were reported. Some patients also exited at this point hoping for better symptomatic management and care from alternative medicine.
Okay, I am always in pain …., I think Ginseng tablets are the best solution………. it takes less time to manage my pain levels…. I myself decided to take it but I asked the doctor if I could keep using it; he said, “You know if you feel good about, no problem using it”—Female, 56 years old, living with peripheral neuropathy for 7 years.
I massage my fingers, close my hands and move my hand in warm water for some time... I move my fingers, put them in warm water and I massage them again so the pain goes—Male, 62 years old, living with peripheral neuropathy for 10 years.

In general, participants who stayed within the national healthcare system continued the cycles of monitoring and repeat prescriptions for medication with very little improvement in their health. Furthermore, some patients were concerned that they knew very little about their disease or its progression, while others felt disappointed that they were not referred for psychological or dietary support or given any information about alternative therapies.
If they (seminars) are available in the national hospital, it will be so helpful and should be given to patients—Male, 43 years old, living with peripheral neuropathy for 4 years.
The doctor prescribed only the medication but never gave any psychological support or referral for psychological counselling—Female, 60 years, living with peripheral neuropathy for 4 years.

Participants described various difficulties, caused by, in their opinion, not receiving satisfactory care. Patients reported a lack of awareness about their disease and the treatments available, which in turn caused anxiety regarding the progress of their disease. They reported staffing and management issues such as a shortage of healthcare professionals that was evidenced by the long waiting times, not only to see the doctor but also the few very busy nursing staff. Participants also mentioned several administrative problems, such as long waiting time for referrals that were exacerbated by the unavailability or loss of medical records. The lack of universal electronic medical records also meant the unnecessary repetition of history taking, examinations and full investigations. Social practices, such as giving gifts and bribes to porters, helped a few people to bypass waiting lists and receive more medical attention and a better quality of care.

Many patients chose to exit the traditional care pathway to try alternative therapies at different points during their patient journey, especially when they felt frustrated waiting for their appointments or when they did not get the expected care from the medical team or relief from symptoms by following the prescribed medications. Many reported trying self-help options such as ignoring the pain, exercising, dietary change (by seeking out a dietician and educational sessions from private hospitals or Dasman Diabetes Centre in the public sector). Alternative therapies were obtained from the hospital for Islamic Medicine in the public sector or private massage parlours functioning independently or as a part of well-known private hospitals such as Dar Al Shifa Hospital. Most people living with peripheral neuropathy wanted to receive further psychological support and education sessions to help them cope better with their condition.

## 4. Discussion

This patient journey map is the first of its kind in the Middle East for peripheral neuropathy, though there have been other patient journey exercises conducted in other therapeutic fields such as irritable bowel syndrome in Dubai, Qatar, Kuwait and Saudi Arabia [26]. This process map depicts the journey experienced by patients from the primary clinics to the national hospital of neurology. A close analysis of the patient journey map and comparison with standard care in the UK [27] identified both similarities and differences in the healthcare pathway. Similar referrals, investigations and pharmacological treatment were identified, however the range and availability of healthcare services differed. The comparison highlighted that in Kuwait highlighted that a much more strategic approach is required in three main areas including non-pharmacological support, organizational systems and medical care.

### 4.1. Psychological Support

There are three factors that play an important role in pain perception—namely psychological, psycho-behavioural and psychosocial components [28,29]. Many participants in this study reported that their pain had influenced their mood, sleep, relationships and functional capacity. This finding is similar to research reported by Hensing et al. [30] in 2007 that showed examples of exaggerated pain and consequences of chronic pain in neuropathic patients, where the touch of a nightdress triggered a massive stimulus, which in a healthy patient would be negligible.

In the UK people living with peripheral neuropathic pain and their families are given information regarding the causes of the disease, treatment and prognosis alongside psychological support via counselling sessions guided by qualified psychologists [31]. Treatment in Kuwait, however, depends on the pharmacological management of symptoms alone. According to World Health Organization (WHO), effective management of patients’ emotional distress has contributed to the success of primary healthcare, by utilizing the tool of reassurance [32], which places emphasis mainly on patient education and counselling. Improvement in this area to help people living with peripheral neuropathy in Kuwait is highly recommended.

### 4.2. Administrative Problems

In Kuwait, waiting times for referrals from the general practitioner in the primary clinic to the doctor in the general hospital took anywhere between 2 and 8 weeks, to the neurologist in the general hospital another 3–6 months and to the neurology specialist/consultant in the national hospital another 3–6 months. In case of loss of medical records, these times were extended to up to 12 weeks, 8 months and 8 months respectively. The above findings regarding long waiting times for referrals in Kuwait are similar to a study conducted in 2011–2012 in rural areas of Iran that identified inadequacies in the government healthcare referral system [33]. The Iranian study highlighted specific issues with the referral system such as a lack of communication between different levels of the system itself. In addition, the study showed that people living with peripheral neuropathy possessed insufficient knowledge of the system, self-referred or bypassed it entirely. The referral system in Kuwait, from the patient’s perspective, could be improved by coordination between different levels of the referral system, strengthening the public sector of the system, increasing public awareness about the referral system and preventing self-referral, similar to those improvements implemented in Iran [33].

There is no centralized electronic medical record system in Kuwait, which leaves physicians with no choice but to hold bulky paper files, most of which are loosely arranged and often lost during referral. Consequently, succeeding specialists prefer to repeat all tests and treatments instead of spending time deciphering the patient history in the file or communicating with the previous physician. In conjunction, patients are asked to repeat their histories at every stage of the journey. This leads to an inability of the treating physician to visualize the treatment of a patient as a whole [34]. Electronic medical records could therefore potentially reduce delays and overall staff workload. Technological advancements and electronic medical records can have a significant impact on a referral system. They can improve quality of care, patient outcomes and safety through improved management, reduction in medication errors and reduction in unnecessary investigations. Furthermore, they can improve communication by phone, email and face-to-face among primary care providers, patients and other providers involved in care [35]. Electronic medical records have been demonstrated to improve efficiencies in workflow through reducing the time required to create charts, improving access to comprehensive patient data, helping to manage prescriptions, improving scheduling of patient appointments and providing remote access to patients’ charts. Electronic medical records capture point-of-care data that informs and improves practice through quality improvement projects, practice-level interventions and informative research [35]. The suggestion of Aij et al. [36] in the Netherlands, for example, that hospitals and healthcare providers should source third party consultants to train management on how to manage and implement solutions, could be a useful recommendation based on the results of this current study. If this was implemented, the patient journey could be redesigned to avoid repetition, remove inconsistencies and create greater standardization across related departments and organizations [37].

### 4.3. Medical Care

Table 3 shows that pharmacological treatment of peripheral neuropathic pain in Kuwait aligns with the UK NICE and IASP guidelines, except that many diabetics are not prescribed carbamazepine (200–400 mg) or gabapentin (300–1200 mg) tablets as first line treatment [27,38]. Furthermore, topical applications such as capsaicin cream, are not available. Tramadol was rarely used to treat people experiencing diabetic peripheral neuropathic pain. However, the availability of these other medications and training for personnel to use and prescribe them would help people manage their peripheral neuropathy more effectively. Guidelines regarding the intervals for assessment and reviewing changes in medication, for example, when to escalate from 1st to 2nd or 3rd line treatment, could improve the follow-up process. New treatments could then be introduced carefully and with the assessment of their efficacy in the Kuwaiti context.

### 4.4. Implications for Research

The main strength of this study is the high level of patient involvement and the wide range of individuals interviewed. Patient perspectives are not routinely explored through research in Kuwait. There are several limitations, however, that have implications for how future research studies in this area are designed and implemented. This study was hampered by the unavailability of a centralized Electronic Medical Records (EMR) system to confirm the diagnosis and treatment being followed in locations or clinics other than the National Hospital for Neurology. Another limitation was the age range of participants which meant younger people, aged 18–34, were not represented so it was not possible to explore how similar their experiences are to those reported here.

The recommendations outlined in this article could become the basis for further qualitative research in the area. For example, observational studies could assess the success of implementation of recommendations and procedures suggested in the treatment of people living with peripheral neuropathy in Kuwait. At the same time, in-depth interviews with individual family members and medical staff could be undertaken to fully explore their problems and issues. A mixed qualitative approach such as observation and in-depth interviews would provide deeper insights and therefore a broader perspective.

### 4.5. Implications for Practice

In terms of current practice, this study highlights people living with peripheral neuropathy in Kuwait report a lack of psychological and psychosocial care. This could be addressed by healthcare providers introducing psychological support via counselling sessions, guided by qualified psychologists, to their standard care. Furthermore, the overall medical system could be enhanced by addressing the identified issues with communication and organization. With regard to communication, perceptions of treatment for peripheral neuropathy could be improved by focusing on the doctor/patient interaction, along with relationships within the multidisciplinary team. Medical professionals could create a culture of mutual respect and cooperation among both medical and administrative colleagues to view patient care as a collective and collaborative effort [39]. Furthermore, in regard to administrative issues, a way to address the current deficit would be to employ a more organized approach to the care pathway, in particular the use of electronic medical records.

## 5. Conclusions

This map of the patient journey of people living with peripheral neuropathy is a fresh representation that Kuwaiti health officials and medical personnel might find helpful in visualizing the process from the patient’s perspective. The study and journey map provide evidence based on interviews that there are several shortcomings and weaknesses in the medical and administrative systems, at all levels (primary and secondary care) in Kuwait, which people living with peripheral neuropathy have to overcome. The process map indicates where changes can be made to improve patients’ experiences and potentially their satisfaction with their healthcare and treatment. By addressing administration, medical care and psychological support issues highlighted in this study, people living with peripheral neuropathy could experience more positive treatment outcomes.

## Figures and Tables

**Figure 1 pharmacy-07-00127-f001:**
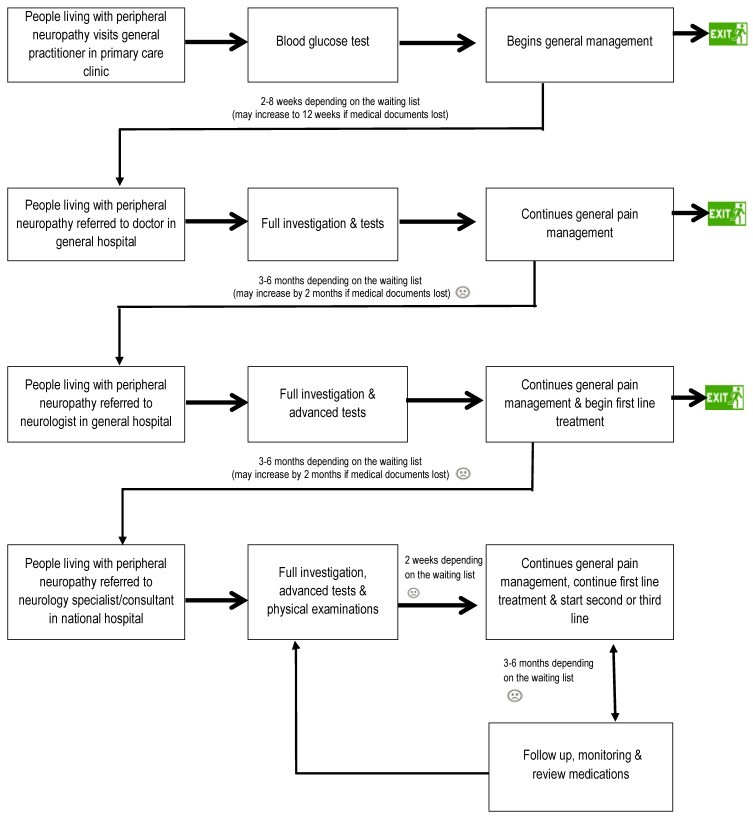
A Process Map of People Living with Peripheral Neuropathy in Kuwait.3.1. Primary care clinic—the journey starts.

**Table 1 pharmacy-07-00127-t001:** Characteristics of people living with peripheral neuropathy in Kuwait.

Characteristics		Observations (*n* = 25)
Nationality	Kuwaiti	20
	Non-Kuwaiti	5
Sex	Male	12
	Female	13
Age (years)	Range	35–82
	Mean (Standard Deviation)	55 (SD = 10)
Comorbidities	Type 2 Diabetes Mellitus	16
	Hypertension	10
	Dyslipidaemia	8
Duration of peripheral neuropathy (years)	Range	3–30
	Mean (Standard Deviation)	13.76 (SD = 7.4)

**Table 2 pharmacy-07-00127-t002:** Investigational procedures carried out for people living with peripheral neuropathy in Kuwait.

Centre	Investigational Procedures Performed
Primary clinic-general practitioner	Blood routine, random blood sugar, fasting blood sugar and postprandial blood sugar for diabetics
General hospital-doctor	Blood routine and HbA1c (glycosylated haemoglobin)
General hospital-neurologist	Blood routine, HbA1c and other blood test, advanced neurological investigations: loss of protective Sense, +1 of reflex ankle-vibration- pin prick- gait
National hospital-neurology specialist/consultant	Physical examinations, full neurovascular examination, nerve conduction study and foot care diabetic peripheral neuropathy. Investigation and examination: HbA1c and other blood tests

**Table 3 pharmacy-07-00127-t003:** Treatment options provided for people living with peripheral neuropathy in Kuwait. * The current study showed that people with peripheral neuropathy, who are mainly diabetics, were not using Carbamazepine 200–400 mg tablets or Gabapentin 300–1200 mg tablets as first line medications.

General pain management	ketoprofen (Fastum 2.5%) Gel	1
	etoricoxib (Arcoxia) 30 mg tablets	1
	ibuprofen (Ibuprofen) 200–400 mg tablets	10
	acetaminophen (Acetaminophen) 500 mg tablets	5
	alpha-lipoic acid (Thiotacid) 600 mg tablets	2
	vitamin b complex (B complex) 500 mg tablets	7
First line treatment *	pregabalin (Lyrica) 75–150 mg tablets	8
	Tricyclic antidepressants (TCAs): Amitriptyline (Tryptizol) 25–150 mg tablets	1
Second line treatment	Selective serotonin and norepinephrine reuptake inhibitor (SSNRI): duloxetine (Cymbalta 60–120 mg capsules and Cymbatex 30 mg capsules)	1
Third line treatment	Opioids: tramadol (Tramol) 50 mg Tablets	1

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
