# Peer review of "The Experiences of People Living with Peripheral Neuropathy in Kuwait—A Process Map of the Patient Journey"

_pharmacy, 2019, doi:10.3390/pharmacy7030127_

Round 1

Reviewer 1 Report

An interesting paper, well presented small scale study.

Author Response

Thank you for your review of our paper.

Reviewer 2 Report

Good day. I truly enjoyed reading your paper. As a clinician, I always try to look at things through the perspective of the patient and their families to help us improve the care we provide, often at their most vulnerable times.

I feel your paper is very well done and offer a few minor suggestions:

In line 114, provide a reference for the NVivo12 software for those who wish to research it further

For the patient comments, perhaps place them into text boxes or bold print to differentiate them from the main body of the paper

In your "Clinical Implications" section, possibly discuss the need for psychological and psychosocial care?

Again, overall I feel your paper was well done and will hopefully help to improve care for this patient population.

Author Response

Thank you for your review of our paper, please find our response to your comments below.

Comment: In line 114, provide a reference for the NVivo12 software for those who wish to research it further
Response: A reference has been added, see line 112 and reference 24.

Comment: For the patient comments, perhaps place them into text boxes or bold print to differentiate them from the main body of the paper.
Response: The paper has been formatted by the journal. The quotes have been indented and put into italics to differentiate them from the main body of the paper.

Comment: In your "Clinical Implications" section, possibly discuss the need for psychological and psychosocial care?
Response: The text has been edited, see line 403-406. The following text has been added:
“In terms of current practice, this study highlights people living with peripheral neuropathy in Kuwait report a lack of psychological and psychosocial care. This could be addressed by healthcare providers introducing psychological support via counselling sessions, guided by qualified psychologists, to their standard care.”

Reviewer 3 Report

Peripheral neuropathy is a painful condition, and required timely and careful treatment for its better management. In this regard, present study by M. Alkandari et al. demonstrates viewpoints from patients and current system of Kuwait to manage this neurological condition. The manuscript is written well and analyzed appropriately. I am wondering if long waiting time to see physician is specific to peripheral neuropathy or this is the case of all other diseases as well. In either case, this study would be helpful for both physician and patients to understand the shortcomings of current medical treatment of peripheral neuropathy.

Author Response

Thank you for your review of our paper.

Reviewer 4 Report

Thank you for the opportunity to review this manuscript. I have a number of suggestions to improve readers’ ability to understand what the authors have done. The comments have been ordered according to the section and page number of the manuscript.

Introduction:

Pg 2, line 63-65 – Can the authors explain the nature of healthcare pathway under investigation in greater detail?

Materials and methods:

Pg 2, line 80-83 - If this observation was informal, I am not sure it should be mentioned later as part of the data, unless data was obtained and recorded? If data was recorded what and how was it recorded?

Pg 3, line 98 – Can the authors provide an outline of how the interview guideline was developed and the questions that it contained?

Pg 3, line 99-100 - For international readers would it be possible to provide some examples of how the healthcare system differs?

Pg 3, line 108-111 - What did these reviews suggest to the authors?

Pg 3, line 114 – If the observational data played a part in the final results please outline how it was analyzed and included with the interviews?

Pg 3, line 121-124 - Are these the general steps in process mapping?

Pg 3, line 126-127 – Can the authors please explain exactly what is meant by “layering on details” from the remaining interviews?

Pg 4, - Should figure 1 be moved to the results section?

Results:

Pg 4-5, line 144-151 – Suggest removing the sentence starting, “Some attended their primary clinic…” through to the quote ending, “…when I am tired and feel bad.” As it doesn’t seem to add to the rest of the discussion.

Pg 7, line 246-247 - To help to contextualize these wait time details would it be possible to have some information about the demographic make-up of the region in which this research was conducted?

Pg 8-9, line 290-292 - How many of the patients had diabetes?

Discussion:

Pg 9 – line 323-325 - What does the comparison to the NICE standards look like?

Pg 111, line 391-392 - There has been no description of the kinds of people interviewed for this study please provide at the beginning of the results section.

Author Response

Dear Reviewer 4

Thank you for taking the time to review our submission. We have addressed the comments made and detailed the changes below. The changes also appear as tracked changes in the document attached.

Introduction:

Pg 2, line 63-65 – Can the authors explain the nature of healthcare pathway under investigation in greater detail?

          Further detail has been added to clarify the aspects of the healthcare pathway that were explored. This section now states:

The main purpose of this research was to explore the experiences of people living with peripheral neuropathy by examining the healthcare pathway in Kuwait from the patient perspective. The healthcare pathway will be examined in terms of referrals, investigations and general management.

Materials and methods:

Pg 2, line 80-83 - If this observation was informal, I am not sure it should be mentioned later as part of the data, unless data was obtained and recorded? If data was recorded what and how was it recorded?

             This has been edited to clarify the informal observations were to familiarise the principal investigator with the healthcare setting rather than being a source of data collection that needs to be recorded. This section now states:

The principal investigator informally observed people living with peripheral neuropathy, that attended the outpatient clinic of Ibn Sina Hospital, to familiarise themselves with the healthcare setting.

Pg 3, line 98 – Can the authors provide an outline of how the interview guideline was developed and the questions that it contained?

          Further detail has been added to describe the interview guideline, this section now states:

A semi-structured interview guide was used to explore people’s experiences. Participants were asked initial demographic questions, including age, sex and nationality, along with questions regarding comorbidities and duration of peripheral neuropathy (see Table 1). The interview proceeded with open-ended questions that began by broadly asking about their pain and then moved on to more specific questions about the healthcare they received in terms of medical treatment, medication and whether they felt anything was missing from their care. The interview guide was developed from the literature, taking into consideration the culture and healthcare system in Kuwait.

Pg 3, line 99-100 - For international readers would it be possible to provide some examples of how the healthcare system differs?

                This has been added and now reads:

The healthcare system in Kuwait is comparable with Western processes in terms of referrals from primary care to hospital settings, however long referral times and a lack of specialist care, such as psychological services, are key differences.

Pg 3, line 108-111 - What did these reviews suggest to the authors?

         Further detail has been added to this section on the reviews:

All reviews confirmed that the transcriptions and translations were accurate and consistent.

Pg 3, line 114 – If the observational data played a part in the final results please outline how it was analyzed and included with the interviews?

          This has been clarified in the methods section as suggested in the previous comment. No formal data was collected to be reported.

Pg 3, line 121-124 - Are these the general steps in process mapping?

            Yes these are the general steps used in process mapping in line with the literature. A reference has been added to signpost the reader to further information on the topic and this has been clarified in the text. See below:

A planned approach for producing a process map was applied to the data to ensure a comprehensive representation of the patient journey [25]. 

Pg 3, line 126-127 – Can the authors please explain exactly what is meant by “layering on details” from the remaining interviews?

             This has been expanded to clarify, as below:

This was expanded into a second version by layering on details from the twenty-one remaining transcripts, which involved adding further detail and clarification to the preliminary process map drafted from the initial four participants.

Pg 4, - Should figure 1 be moved to the results section?

                This has been moved.

Results:

Pg 4-5, line 144-151 – Suggest removing the sentence starting, “Some attended their primary clinic…” through to the quote ending, “…when I am tired and feel bad.” As it doesn’t seem to add to the rest of the discussion.

                This has been removed as suggested.

Pg 7, line 246-247 - To help to contextualize these wait time details would it be possible to have some information about the demographic make-up of the region in which this research was conducted?

           Additional information on the region has been added and this section now states:

If deemed necessary, people may also be referred by the doctor in the general hospital to a neurologist in the same hospital. There are only five general hospitals that provide healthcare to approximately 4 million people living in Kuwait.

Pg 8-9, line 290-292 - How many of the patients had diabetes?

        This information is included in the participants characteristics table (see Table 1) and has been added to this section in relation to the diabetic peripheral neuropathy. This now states:

Nerve conduction studies and advice on podiatry care for people with diabetic peripheral neuropathy were also provided here (16 of the 25 people interviewed, had type 2 diabetes mellitus).

Discussion:

Pg 9 – line 323-325 - What does the comparison to the NICE standards look like?

              This section has been expanded to clarify the comparison, as described below:

A close analysis of the patient journey map and comparison with standard care in the UK [26] identified both similarities and differences in the healthcare pathway. Similar referrals, investigations and pharmacological treatment were identified, however the range and availability of healthcare services differed. The comparison highlighted that in Kuwait a much more strategic approach is required in three main areas including non-pharmacological support, organizational systems and medical care. 

Pg 111, line 391-392 - There has been no description of the kinds of people interviewed for this study please provide at the beginning of the results section.

                   Further detail and signposting to the table with the participants characteristics has been added. This has been added to the results section:

The results were derived from interviews with people attending the outpatient clinic of Ibn Sina Hospital. Most participants were Kuwaiti (n=20), with an average age of 55 years (SD=10) and had been living with peripheral neuropathy for an average of 14 years (SD=7). Further details of the participant characteristics can be found in table 1.

Thank you again for reviewing our submission and we feel the changes made have improved this piece of work.

Kind regards

Round 2

Reviewer 4 Report

Thank you for taking the time to carefully consider my previous comments. I believe they have been adequately addressed.